## Theories

aesthetics; biodiversity; color patterning; morphogenesis.

**Corresponding author:**
Pierre Galipot;
Email: pierregalipotpro@gmail.com

**Associate Editor:**
Daphné Autran (No special collection)

# Colour pattern studies: the SE (せ) method, a shape-centred approach to explore biodiversity and avoid aesthetic biases

Pierre Galipot

UMR CNRS 6553 ECOBIO, Rennes, France

**Abstract**

The preference towards colourful patterns generates many aesthetic biases, including in Biology research, leading to taxonomic preferences and understudied groups, including many plant taxa. After reviewing the importance of aesthetics in Turing colour pattern studies and the relative nature of the sense of beauty in Biology, I present a method called SE (せ) that strongly reduces taxonomic preferences in colour pattern formation studies, together with allowing the exploration of colour patterns biodiversity and facilitating the discovery of new morphogenesis processes.

## 1. Introduction

In colour patterning, morphogenesis (the study of shape formation) and other studies related to biological shapes and colours, aesthetic biases might be stronger than in the majority of other domains of research, because visual aesthetics are a major source of taxonomic preferences (Adamo et al., 2021; Dixon, 2021; Thompson & Birkhead, 2020; Troudet et al., 2017).

Personally, and like many colleagues, I might have chosen this field of study notably according to aesthetical considerations, partly consciously and partly unconsciously. This is the first aesthetic bias I have faced in my young career, and then I try to be careful not to add others in my work. To do that, I try to centre my studies not on biological species but on shape themselves, and for every shape (for instance, repeated patterns, physical cracking patterns or checkerboard patterns etc.), I start the studies by the most exhaustive possible survey thanks to museum specimens, online databases and other sources of data. Then, I chose one or several species to zoom into the morphogenetic processes, with a preference of plant species, when functional experiments are required to completely decipher the mechanisms.

The objective of biodiversity exhaustivity in shape studies permits not only to strongly reduce taxonomic preferences but also to make new discoveries, including deciphering new morphogenetic processes and ways to produce a particular shape. Furthermore, it permits to compare very distant species and to pinpoint evolutionary convergences concerning phenotypes (for instance, colour patterns) and/or processes (i.e., mechanism). Although a consensus on precise common definitions is probably still pending in the morphogenetic field (one of the current works of a collaborative group of morphogeneticians, who started discussions at EuroEvoDevo 2024), I define here a morphogenetic mechanism as follows: a morphogenetic mechanism is a category defined by the set of laws governing a system (whether of molecules, tissues or organs) that are necessary and sufficient to generate the shape of interest regardless of the physico-chemical nature and scale of its agents).

Here, after a quick survey of the importance of aesthetics in colour patterning studies and a discussion about the changing nature of our aesthetic perception of a species during a study, I will then present and outline the main principles of the method I try to apply, named SE (せ), and its benefits.

### 1.1. Colour patterning and the importance of aesthetics

By using Turing colour patterns as an example, we can trace the influence of aesthetics in the history of the field, in particular for the choice of model species.

In our knowledge, the Turing mechanism – initially based on reaction-diffusion and theorised by Alan Turing in 1952 (Turing, 1990, republished in 1990) – has been deciphered or extensively explored for colour patterns in no more than three species or genera (the zebrafish(es) *Danio*, the flowers of *Mimulus*, and the domestic cat), and start to be explored in the lizards and snakes (Milinkovitch et al., 2023). First, *Danio rerio*, the zebrafish, which was extensively studied by Kondo et al. (2021), and by Pr. Christian Nüsslein-Volhard (for instance, Irion et al., 2016). In the 90s, Shigeru Kondo was about to start a career in another domain than colour patterning when the observation of colour pattern changes during the growth of his *Pomacanthus* (a common species kept by aquarians) led him to bifurcate his research interests to found the modern biology of Turing patterns (Kondo & Asai, 1995).

Both *Pomacanthus* and *Danio rerio* were appreciated by the aquarists for their beauty, so aesthetics are at least partly responsible for their presence in biological studies (Rabinowitz & Myerson, 1966; Stanton, 1965).

According to Mullins et al. (2021), Pr. Nüsslein-Volhard's team article on zebrafish colour pattern has been one of the few (among 37) not accepted in the special issue of Development in 1996, presenting the results of the molecular screening of zebrafish by four teams. Her thought is that at that time in the developmental biology community, colour patterns research might not be considered sufficiently related to development to be published among other papers (Nüsslein-Volhard, 2012), and furthermore, their visual beauty might have acted as an aggravating circumstance to be rejected, because of its alleged superficiality (as she told in a conference at the École Normale Supérieure of Paris, in 2018). Now, my feeling is that this snobbism against colour patterns has almost completely vanished, as many research teams, from developmental biology to ecology focus on them.

Concerning plant colour patterning, processes for *Mimulus* flower spot patterns have been deciphered recently by Yaowu Yuan and colleagues (Ding et al., 2020). Most flowers and their colour patterns are seen as aesthetically positive and *Mimulus* are no exceptions: they are colourful (from yellow to red and pink through orange shades) and patterned (notably spots). The centre of diversification is found in North America, and *Mimulus* have genetically and evolutionary been studied by local teams in the 90s and then became a model genera (Yuan et al., 2019).

Finally, Kaelin and colleagues explore the basis of mammalian coat patterning with the cat as a model species (Kaelin et al., 2021). Initially, Kaelin was interested in Agouti-Related Protein and genetics in mice, and came to colour patterning by the role of the related Agouti protein (ASIP) in colour patterning in mammals. As laboratory mice and rats do not exhibit very interesting colour patterns, the lab investigated cats (and dogs) as candidates. It is not to be demonstrated anymore that these domesticated human companion species are selected notably according to aesthetic considerations.

## 1.2. The very changing nature of aesthetic judgments of species of interest

Our judgment of the beauty of a particular species is evidently dependent on many parameters, and as scientists spending a long time in contact with a species, we may change our taste preferences. To illustrate the changing nature of our aesthetic judgments, I will choose several personal examples.

To begin with a short anecdote, I have not been able to recognise the beauties of the common poplar tree. However, I have recently changed my perception after after sessions of *in vivo* leaf morphometry with a colleague.

Second, I have recently experienced another example of the complexity of aesthetics with my students. It is commonly accepted that beauty can arise when changing the scale of observation (we might think of the tremendous amount of SEM electronographies generating visual satisfaction on the observers). Less evidently, it could also be the same for colour patterning. When confronted with two trees presenting little rounded and coloured fruits, *Viburnum tinus* and *Callicarpa* sp. (beautiful fruit in Greek), most people would choose the latter one as their favourite in terms of visual aesthetics, probably due to their shininess and their bright pink colour, contrary to *Viburnum* fruits that are more discrete (Figure 1, upper panels). Surprisingly, when peeled and observed under the stereomicroscope, the epicarp tissues bearing the coloured cells trigger an inverted preference. If *Callicarpa* colouration is based on the common anthocyanin pigments of flowers, *Viburnum* blue colour relies on ordered lipid droplets generating physical colouration (Middleton et al., 2020), leading the cells to look like tiny iridescent crystals (Figure 1, lower panels).

Finally, I recently reconsidered my aesthetic judgment on some common flies called *Sarcophaga*, known as common flesh flies. Most people have a negative view of them (Rozin & Fallon, 1987), partly linked to their behaviour (their larvae feed on flesh), to the cultural disgust towards insects and their grey body colour.

After my PhD supervisor Florian Jabbour laid my attention to one of its favourite plant species, *Fritillaria meleagris*, famous for its unusual checkerboard pattern on its tepals (Figure 2a), I started to be interested in these rare patterns. A biological survey in Eukaryotes brought me to build a list of species bearing more or less defined checkerboard patterns, and among them many *Sarcophaga* species. As well as being among the most geometrically well-defined checkerboard patterns of all species we found, I was surprised to discover that the black-and-white pattern they bear on their abdomen changes according to lighting orientation and viewing angle. Black squares become white and vice versa (Figure 2b). This very surprising feature led me to choose this species, along with the aforementioned *Fritillaria*, to study the formation of checkerboard patterns in Eukaryotes (Galipot & Zalko, 2024, preprint). After spending a few months studying them, my aesthetic taste of these flies has changed to a positive perception, and I will never look at them in the same way again.

Without the preliminary research of checkerboard patterns among Eukaryotes, I would probably not have considered these flies, and would have missed these incredible patterns. This is one of the reasons why I now recognised the importance of exploring biodiversity in morphogenetic studies, and this led me to build and apply a method that I am now going to detail a little more.

## 1.3. The SE (せ), Search for Exhaustivity method to study biological shapes

Whether kanjis (Sino-Japanese ideograms), hiraganas or katakanas (Japanese syllabics), Japanese characters are written using strokes in a precise order and direction, inherited from calligraphy. For the hiragana せ [se], the horizontal stroke is the first to be drawn, followed by the small and large vertical strokes that become horizontals at their ends (Figure 3, left).

I used this character as a visual analogy for the morphogenetic approach I am proposing, named '**SE**' method, which also serves as an acronym for 'Search for Exhaustivity', one of its main principles.

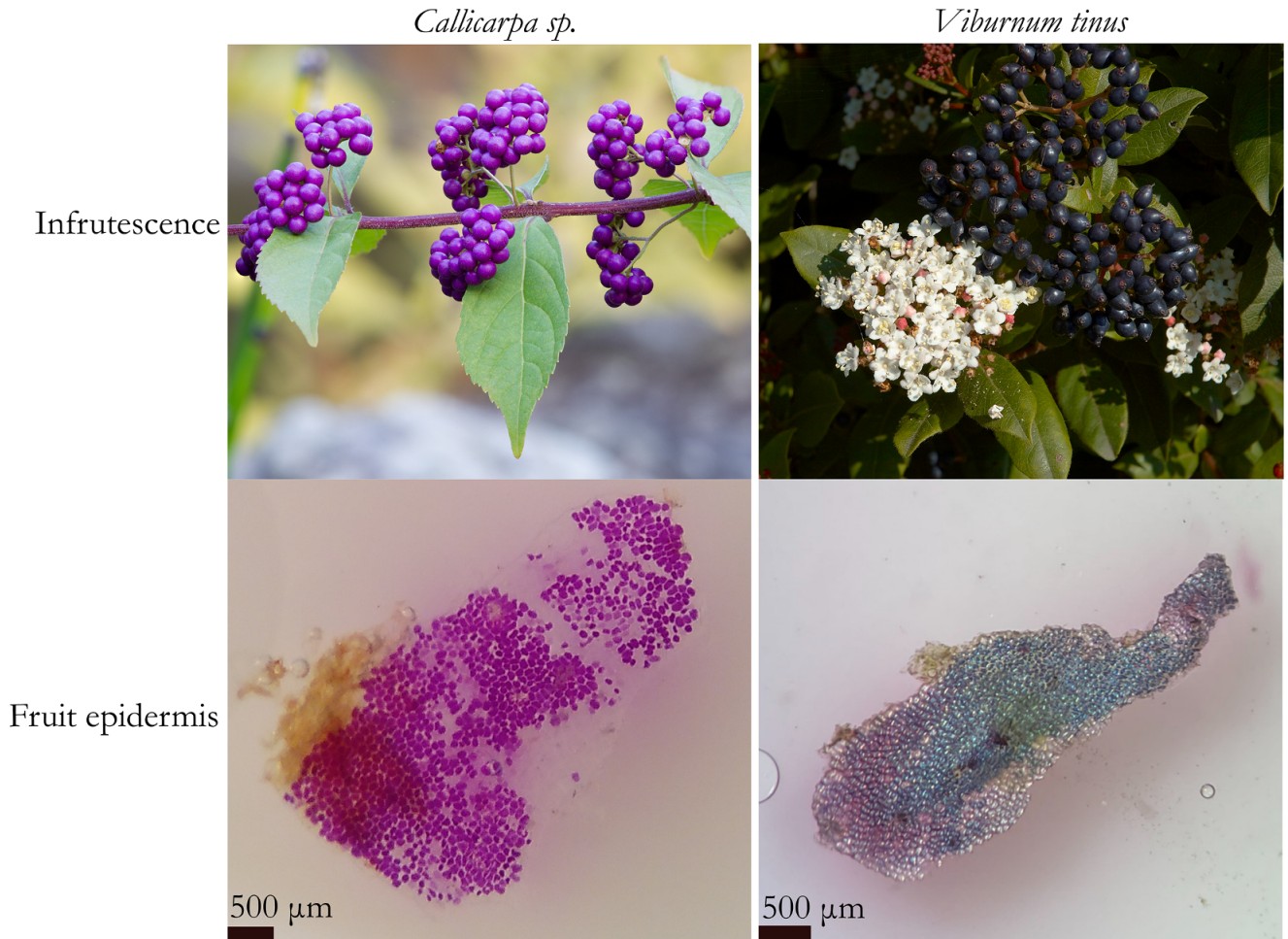

**Figure 1.** Macroscopic (upper panels) and microscopic (lower panels) views of *Callicarpa sp.* and *Viburnum tinus* fruits. Upper panels picture sources: https://commons.wikimedia.org/wiki/File:Purple_beautyberry,_October_2015_-_Stacking.jpg+, https://commons.wikimedia.org/wiki/File:Viburnum_tinus_C.jpg. The fruits of *Callicarpa sp.* and *Viburnum tinus* were collected in the Université de Rennes Beaulieu campus gardens, dissected in distilled water and observed under a stereomicroscope at a magnification of ×40.

The main steps are as follows:

**1.3.1. Step 1.** After choosing a shape of interest (e.g., checkerboard patterns), I start with a horizontal exploration of pattern diversity, by finding as many species as possible presenting these patterns and described the latter. This could require the use of multiple sources, including non-academic ones such as image browsers (e.g., Google Images); wikis (e.g., Wikimedia Commons and Wikipedia) or semi-professional amateurs (e.g., www.orchidspecies.com by Jay Pfahl and colleagues). This could compensate the lack of scientific literature, or the impossibility to efficiently navigate sources and process information. Then ideally, every data should be double-checked by peer-reviewed academic sources. If these are not available (or require an unreasonable time of human processing), a proper citation of the source (mandatory) and the (non-mandatory) possibility for the database to be open and collaborative (like the Gephebase compiling genotype-phenotype relationships (Courtier-Orgogozo et al., 2020) would improve your database and the conclusions you draw from it. Some studies try to rationalise risk assessment (Forstmeier et al., 2017), that is, to avoid errors (false positives and negatives, unbalanced representation of taxonomic diversity etc.) but still are often too generalist to be able to provide tailored advice for your particular database.

Concerning the representativity issue, as the extend of the natural biodiversity often prevents a complete exhaustivity and because a look at every discovered species would take too much time for the researcher, strategics choices are required. A possible way is to choose a given taxonomic rank (e.g., orders for Animalia), and to explore every taxon at this taxonomic rank. An unbiased random selection of species could be another solution, and many papers – notably coming from Biology of Conservation studies – propose techniques that would guide you to the more appropriate way to reach representativity (e.g., Sastre & Lobo, 2009). Then, of course data might not be available for every species, and generate a new bias, possibly in the favour of beautiful species (see Section 1). Nevertheless, a big advantage of colour pattern studies is that photographs or at least text description of the exterior colour patterns of the species is very rarely missing, as an account of the optical appearance of the species is one of the very first steps of species description (see Braby et al., 2024; Rainey, 2011; Seifert & Rossman, 2010).

In the Putative Growth Turing Colour Patterns (PGTCPs) project, to reach a satisfying level of exhaustivity compatible with the time allowed for a research project (for this particular one, 4 months full-time were required to make the survey), I applied a method adapted to the project and the number of species

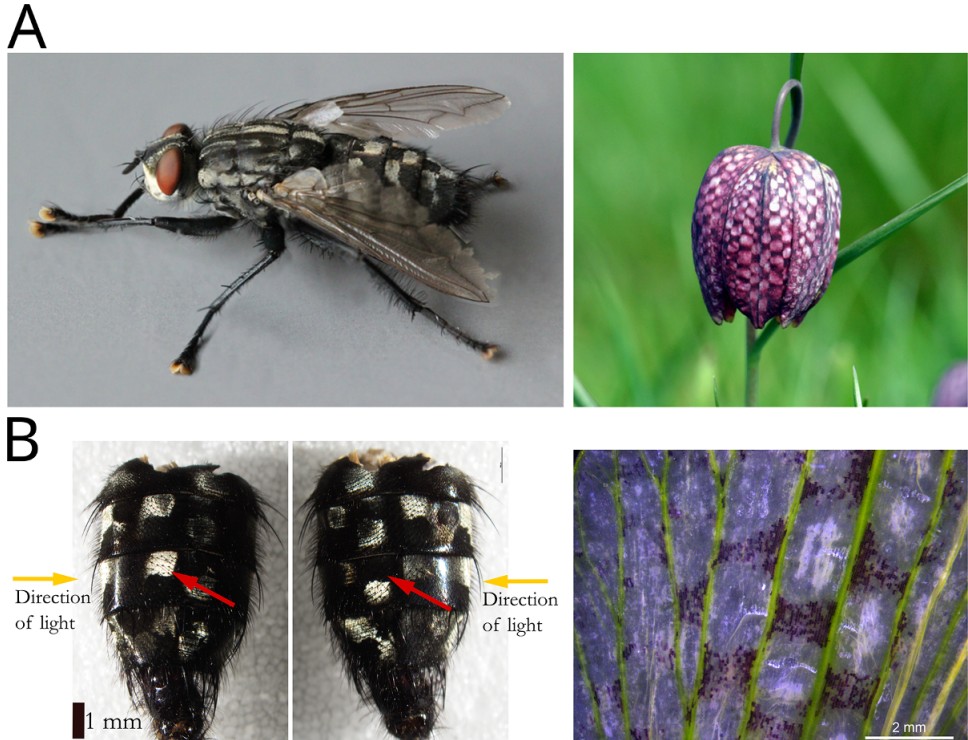

**Figure 2.** Checkerboard patterns of *Sarcophaga sp.* and *Fritillaria sp.* (a) *Sarcophaga* presents checkerboard patterns on its abdomen, *Fritillaria* on its sepals and petals. Image source: https://commons.wikimedia.org/wiki/File:Fritillaria_meleagris_14.jpg, https://commons.wikimedia.org/wiki/File:Sarcophaga_variagata,_Gop_Hill,_North_Wales,_July_2013_%2817168726209%29.jpg. (b) Left: Detail of *Sarcophaga sp.* abdomen. Squares change their colours depending the angle of light and the angle of observation. This structural (= physical) colouration seems driven by the length and directionality of abdomen bristles. Right: Detail of *Fritillaria thunbergii* petal pattern. Cells are probably coloured by anthocyanin pigments in their vacuoles. Fruit epicarp and tepal observations. The flowers of *Fritillaria sp.* in Figure 2b were collected in the Botanical Gardens of the University of Tokyo, dissected in distilled water and observed under a stereomicroscope at a magnification of ×20.

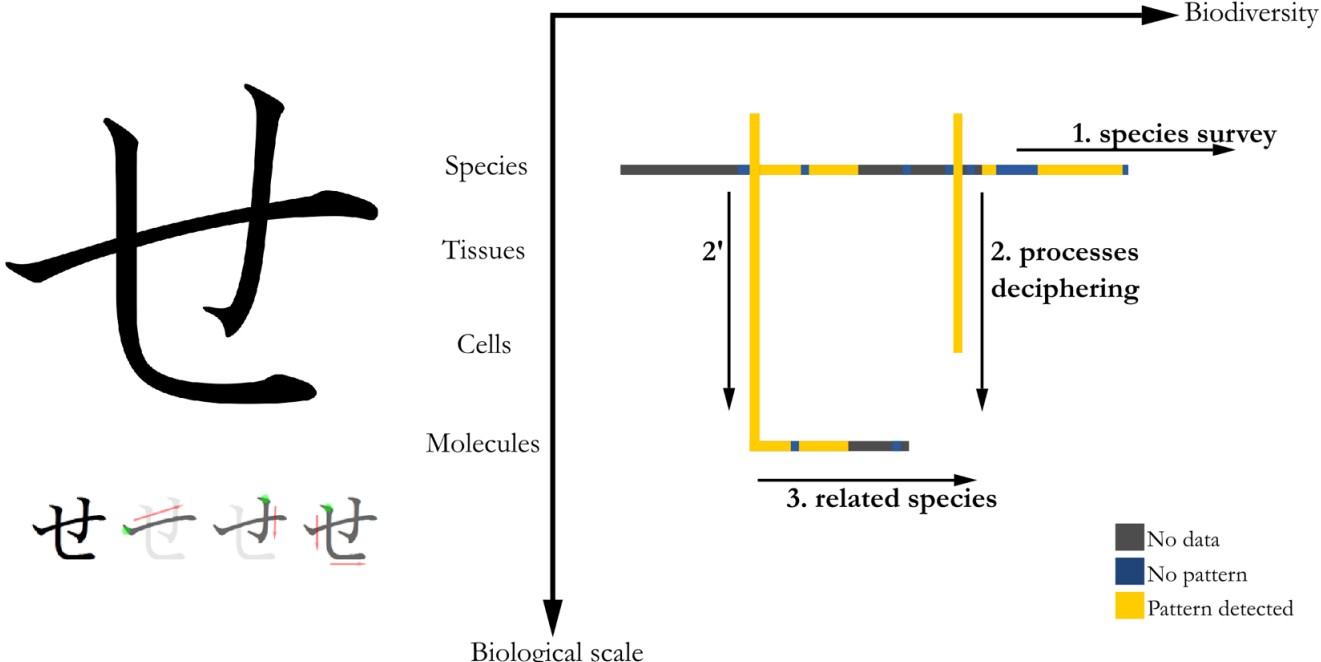

**Figure 3.** Left: Japanese hiragana character corresponding to [se] sound, and its orderly way of writing, serving as a visual analogy to the method. Right: The three steps of the SE method represented in a Biodiversity VS Biological scale graph. First step: species survey; second step: processes deciphering by zoom into tissue, cells and/or molecules scales and third step: use of related species to help deciphering pattern processes.

(represented under a flow chart in supplementary figure 3B of Galipot, 2025).

**1.3.2. Step 2.** Following this preliminary study, I choose ideally two or more appropriate species to study the ontogeny of these patterns, that is, a vertical study of morphogenesis. Although there is no universal definition of what is the best species for processes deciphering, here are my personal criteria, additional to the classical criteria (easy to breed/grow, short life cycle, relatively small individuals, genetic or molecular tools eventually available etc. (Jenner & Wills, 2007):

- Choose some phylogenetically distant species, including at least one outside the most studied group in colour pattern studies – animals – in order to study the potential evolutionary convergences and to propose an interesting discussion on homology or analogy of the studied trait.
- Choose species presenting a very clear pattern (in terms of geometrics, of motif number on a single individual etc.), present in a large proportion of the population, but ideally facing some inter-individual polymorphism (or alternatively variation in closely-related species). This will allow to facilitate the reproducibility of the experiments in order to reach an appropriate sample size and in the same time to use variation to better understand the processes (see examples of non-mandatory Step 3 in the next section).
- Choose species presenting a pattern compatible with the planned experimental design (e.g., on a tissue accessible at the right developmental time). To determine this, it might of course require preliminary studies or literature exploration. The timing of pattern formation could be crucial; if patterns only appear very early in development or in hardly accessible tissue. For colour patterning, it is very common that pattern formation takes place before colour production, therefore being not directly accessible to the eyes or classical observation instruments (Galipot, 2025).

**1.3.3. Step 3.** (Non-mandatory but often informative), I use related or outgroup species (a return to horizontality) to help decipher the patterning processes and their recent evolutionary history, which is a corollary of the founding principles of evo-devo (the study of the evolution of developmental mechanisms, Richardson, 2022).

The biodiversity approach in these first steps should bring three advantages:

First, it increases our knowledge of biodiversity, here from the angle of particular traits, colour patterns. This work can join one of the missions of museums of natural history and if the database is well organised and accessible, data can be exploitable for future studies of evolutionary biology, ecology and various other fields of research.

Second, the search for exhaustivity permits to reveal novel patterns in understudied species. It is of course completely dependent on data availability. A species lacking pictures, drawings or precise descriptions will be out of the scope, but still, it permits to avoid many biases, including the aesthetic ones evoked earlier. The more species are described with the pattern of interest, the more comfortable will be the choice of species to study pattern formation, thanks to a large range of candidates.

Third, this permits the choice of very distant species (typically, a plant/a fungi and an animal) to study pattern formation in two taxa in parallel, and could bring useful information for the presence or absence of evolutionary convergences. If evolution considerations are not a direct objective of the SE method, the latter may generate informative data which can be used in upcoming studies.

## 1.4. Examples of application of the SE (せ) method

In this section, I illustrate and comment each step of the method with previous or ongoing works.

**1.4.1. Step 1: Biodiversity survey.** The complete species survey I have produced was done to study of a particular category of periodic colour patterns, the PGTCPs. They consist of classic Turing patterns whose shapes and/or colours are disrupted by tissue growth, the most famous example being leopard rosettes (Galipot, 2025). Before this study, they were described in only a few animal species (leopard, a dendrobates frog, a rodent and a whale shark), but not in any plant species. To increase our knowledge about these colour patterns, I tried to find at least one example of PGTCPs in every animal order, and every Angiosperm family. Then, I made an exhaustive survey of orchid and mammalian species, two groups where photographic or drawing data were available for every described species. The results were beyond expectation quantitatively and qualitatively, because not only many diverse species all around the Eukaryotic phylogeny were found bearing these particular patterns, but also new varieties of PGTCPs (in terms of shape) were described, notably in orchid, sea slug and cephalopod species. This amount of new data allows to build strong hypotheses for putative biological functions of PGTCPs (in a nutshell, being more and less visible) and to pinpoint many evolutionary convergences.

For the checkerboard pattern study (Galipot & Zalko, 2024, preprint), even if the biodiversity survey was not as exhaustive as the precedent example, it permits to find very diverse examples of this quite rare pattern: Actinopterygii, snakes and lizards, cephalopods and flies species joined the Liliacea *Fritillaria meleagris*, the starting point of the study.

**1.4.2. Step 2: Pattern formation deciphering.** The PGTCPs study allows us to find the best species of interest to perform functional experiments. From the four or five animal species described before the survey, we could make a choice between thousands of species, including more than 300 orchids (Galipot, 2025). Finally, we selected a Lamiaceae plant, *Scutellaria rubropunctata*, known for its purple rosettes on its petals (Fujita et al., 2015). This plant met almost all of our criteria (Galipot and Tsukaya, in prep).

For the checkerboard patterns, like evoked before, it permits to add the flies from *Sarcophaga* genus to the study, and the comparison between their pattern morphogenesis to the *Fritillaria* ones showed two completely opposite ways to produce alternate squares, a perfect example of convergence of shapes but not of processes. Briefly, in *Fritillaria*, Turing pattern dots are transformed into squares by geometric constraining, while in *Sarcophaga*, two orthogonal groups of stripes combined to produce checkerboard patterns (Galipot & Zalko, 2024, preprint).

**1.4.3. Step 3: The use of related species to reveal patterning processes.** In the case of our study on checkerboard patterns (see Galipot & Zalko, 2024 and Figure 2), even if the biological survey was not as exhaustive as the precedent example, Step 1 of the SE method permits to find very diverse examples of this quite rare pattern. Pattern variations among the *Fritillaria* and *Sarcophaga* genera, together with the use of outgroups (notably the sister genus *Lilium*), were crucial to suggest processes of pattern formation

and to bring indirect proofs. More precisely, all intermediaries between dots and squares are present in the *Fritillaria* genus, showing that checkerboard patterns could be modified classic Turing spot patterns, like those bared by the sister genus *Lilium*. The observation of rare individuals of *Fritillaria meleagris* presenting spot patterns instead of checks, together with the tremendous diversity of patterns present in the other species of the genus, allowed us to strongly suggest a role of the parallel venation network in the formation of the pattern.

In the PGCTPs project, together with the use of related species, I added multi-scale comparisons: intraspecific and intra-individual (between tissues or developmental stages) to build a cluster of clues pointing to the role of growth in PGTCPs formation.

## 2. Conclusion

### 2.1. Deep-learning-assisted studies are the future of SE (せ) method

Although SE (せ) method seems to provide promising results in colour pattern analyses, it is a time-consuming workflow. As an example, the PGTCP study biodiversity survey took me between 3 and 4 months, full-time and only two groups were almost exhaustively analysed (orchids and mammals). Furthermore, the relative abundance of PGTCPs among animals and angiosperms accelerated the search, because the project was to find at least one example of species for each animal order and each plant family, so I could switch to the next clade as soon as an example was found. If I had wanted to apply the same technique to checkerboard patterns, it might have taken a year or more. Instead, I leveraged the fact that checkerboard patterns are rare in nature and visually appealing. This led many scientists to mention and describe them during the descriptions of the concerned species in their articles, so I use a mix of keyword-based searches to species-by-species exploration, but still, this survey is far from exhaustivity.

To solve this problem, the most promising tool for the next few years will be artificial intelligence (AI) and especially deep learning networks, which are already extremely powerful in image analysis and shape recognition. If typically there is a need for 10000 images to train a deep-learning network (Lhermitte et al., 2022) and therefore could be incompatible with the amount of species available before the study, this could be solved by using both real and simulated patterns to train the network. If every species where a pattern is detected by the AI tools would be systematically verified by a human, it could greatly accelerate studies and make them many times more exhaustive than what humans could do. It could then contribute to build a large, open-access colour pattern database centred on shapes, which could serve for basic knowledge and biodiversity descriptions, but also for evolutionary and ecology studies, as well as helping functional studies to decipher processes of pattern formation.

### 2.2. From natural patterns to human-made patterns

Trying to reduce aesthetic biases does not mean to completely exclude aesthetic considerations in my work, just being more aware of it. In an upcoming project, I would like to try to apply the SE (せ) method to very particular colour patterns, the human-made ones and particularly in artistic productions. In collaboration with artists and specialists on the basis of aesthetics, I would like to perform pattern analyses to understand the links between micro-patterns (the local techniques of drawing)

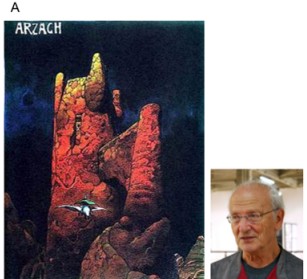
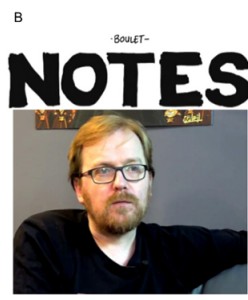

**Figure 4.** (a) Low-resolution version of an illustration of Arzach comic cover, considered of one of the masterpieces of Jean Giraud/Moebius, and a photograph of him. (b) Title drawing of the first comic of the series 'Notes' published by Gilles Roussel/Boulet, containing many strips drawn with (or talking about) the 'petits-traits' technique, and a photograph of him. Images sources, from left to right, up to bottom: https://upload.wikimedia.org/wikipedia/commons/5/56/Moebius_Lodz_2008.jpg, https://upload.wikimedia.org/wikipedia/en/f/f0/Arzach.jpg, https://upload.wikimedia.org/wikipedia/fr/3/36/Notes_de_Boulet.png, https://upload.wikimedia.org/wikipedia/commons/2/2a/Boulet_mollat2017.jpg.

and macro-patterns (the overall drawing and its aesthetics perception). A starting point might be the very famous technique of 'petits-traits' (small lines), theorised by the comic cartoonist Boulet (Gilles Roussel, notably in its website/blog 'Bouletcorp' https://www.bouletcorp.com/notes, see Figure 4b) and used in comics and manga with many variations. Although value judgments in art and comparisons between artists are areas to be considered with caution, I would like to shed some light on why so many artists consider Jean Giraud/Moebius (see the official website in French https://www.moebius.fr/index.html, and the Wikipedia page in English https://en.wikipedia.org/wiki/Jean_Giraud) to be the greatest in his field, and in particular what are the particularities of his mastery of 'petits-traits' (see Figure 4a).

**Data availability statement.** All information in contained within the text.

## Acknowledgements

The author would like to thank Olivier Hamant for suggesting him to think about the links between aesthetics and his work in colour patterning. The author deeply thanks Florian Jabbour for drawing his attention to the beautiful *Fritillaria meleagris* flowers which gave him the idea to explore checkerboard patterns, and Julie Zalko for her help on this checkerboard patterns study. The author deeply thanks Agnès Schermann-Legionnet for drawing his attention to the *Populus* leaves which gave him the idea to discuss the relative notion of beauty perception in biology studies. The author would like to thank Hélène Galipot, Robin Barry and Lidia Favaretto for the discussions about Jean Giraud/Moebius works and Lucas Galipot for the discussions about Gilles Roussel/Boulet works. The author deeply thanks the two reviewers and the editors who taking the necessary time and effort to review the manuscript. The author sincerely appreciates all their valuable comments and suggestions, which greatly helped him in improving the quality of the manuscript. In particular, the author deeply appreciates the time they took to read the related articles and preprints.

**Author contributions.** P.G. conceived and designed the study and wrote the article.

**Funding statement.** This research received no specific grant from any funding agency, commercial or not-for-profit sectors.

**Competing interest.** The author declares none.

**Open peer review.** To view the open peer review materials for this article, please visit http://doi.org/10.1017/qpb.2025.8.

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
