## [Reviewer Report]

In this “Theories” manuscript, the author tells a personal scientific and conceptual journey; asking the question of potential subjective biases in choosing a biological object to study color patterns or morphogenesis, due to aesthetic attraction (or repulsion); and proposing a method to control such bias and to build more objective and exhaustive surveys of species of interest, and decipher the underlying mechanisms.

I enjoyed reading the manuscript, for the first person and natural writing, and the different thread of thinking proposed, including the question of subjectivity; and the meaningful, elegantly called, “hiragana”(SE) morphogenetic method; which provides, via simple rules, a way to deepen the scope of the important biological questions of patterning and morphogenesis, by exploring phylogeny. Such endeavor appears timely, at the “post-model species era”, and for biodiversity studies.

In the conclusion, the author opens perspectives on 1) the use of patterns recognition AI tools to extend the power of the method, and 2) applying the method to study art works to ask the question of patterns and aesthetics at multiple scales. These perspectives are very interesting, in terms of developmental and evolutionary biology; and of epistemology and art-science actions, respectively.

In conclusion, this manuscript provides food for thought, nicely fits the scope of the “Theories” format of QPB, and engage the reader to look forward reading the future works in preparation mentioned in the text.

Minor points:

- Figure 2B: the picture at right is not mentioned in the legend. I guess it is a closer view of Frittilaria tepal? Does the pattern also depend on light direction in this specie, as in the fly?

- Line 30: « …concerning phenotypes (patterns) and/or mechanisms (morphogenesis).”: The vocabulary might need some clarification here, indeed both “phenotype” and “mechanisms” are both general terms which can relate both to either patterns or morphogenesis. Could it be rephrased, for instance, as “… concerning phenotypes and/or mechanisms of patterning and/or morphogenesis.”?

- Line 50: after “… species or genera.”, it would help clarifying/structuring the text to list the 3 current model species for color patterns: Danio rerio, Mimulus, and cats, which are described in the next paragraphs.

- Line 84: the sentence on the author’s experience of changing his vision on poplar leaves appears a bit short and isolated, and might be separated from the introductory sentence (“Our judgment…”) and extended to a short paragraph (with introduction and conclusion or transition with the next paragraph).

- Line 216: “in prep” is repeated.

- Lines 247 and 252 : please explicit abbreviation IA, and change to “AI” in second occurrence.

- Line 270: Since the « Material and methods » section is very succinct, it might be integrated in the respective figures’ legends.

- As a suggestion, a concluding figure illustrating the “petits traits” pattern could nicely explicit the proposed idea (ie, the link between micro-patterns and macro-patterns and aesthetics perception)? This would “loop” the article on aesthetics - however I don’t know if “open access” images of this author are available.

- I was curious about the technique "of “petits-traits” (small lines), theorized by the comic cartoonist Boulet (Gilles Roussel)14 “. But in reference 14, I could not find the information (it seems rather a general review on cartoons' web publishing).

- Please review the formatting of the references (notably #9).

---

## [Reviewer Report]

The presented study addresses a rather transdisciplinary subject, i.e. that aesthetic biases in colour patterns of plants and animal might affect biological research through “taxonomic preferences”. The authors suggest a new method (SE) evidencing and, thus, reducing biases in study priority and in this way fostering discoveries of new mechanisms.

While I found the fundamental principles in this approach rather convincing, in the end I was mainly lost with the manuscript. From its structure, it is presented as a combination of a research proposal surrounding an interesting idea/method, presenting unclear results (but interesting biological phenomena). In the end, the reader is not able to follow and, thus, repeat the considerations as there are no results/analyses included in the study. While the main point of the manuscript is the relevance of the SE method, providing a theoretical framework, data and clear examples are missing. Otherwise, the reader is not able to follow the potential theoretical impact of the method presented.

I found part of the text confusing and not very focused on the main aim – I guess that is to invent/present/promote an unbiased view on colour patterns in nature and avoiding aesthetic biases. I also missed an introduction, why such biases might be disadvantageous in research. Only if the scientific community is presently “wasting time” as aesthetic biases channel our research such a method would be very important for unbiased research questions.

I found the use of terms also hard the follow – what is exactly meant by “mechanism” or “morphogenesis” – it is very necessary to define these terms in such a theoretical context.

The examples given I found interesting as “colour” and “pattern” is a highly important evolutionary driver and is depending on physiology, but also morphological structure of surfaces. The fruit epidermis photos and all others would need a scale/size bar. This is scientific standard.

Although I tried I could not fully follow the SE method (Figure 3) and many statements as no data are included in the study and the method is comparatively presented for the surveys of the author.

In summary, while I can imagine that the SE method might have its value, as the article is presented I doubt that is will have any impact as the reader cannot repeat the considerations.

---

## [Reviewer Report]

Review for Manuscript ID QPB-2024-0031

The manuscript “Color pattern studies and aesthetic biases: the SE (せ) method to avoid them” by Galipot addresses the issue of bias in the choice of species for studies of colour patterns, and particularly a bias by aesthetic choice. The author recounts his own experience on selecting study species for a Turing colour pattern study in Eukaryotes.

A method is then presented, the “Search for Exhaustivity method” (SE), which aims to reduce the bias in the selection of possible study species. The steps of the method are: 1) selection of a shape of interest, 2) selection of study species from a preliminary study, 3) (optional) selection of an outgroup species.

The article was submitted to the theories section.

Developing rigorous and quantifiable methods to avoid bias in the selection of study organisms is in my opinion a worthwhile endeavor. It might aid the investigation of structural and/or molecular basis of biological processes, like pattern formation. Moreover, it could help to unravel the question if similarities in processes are due to evolutionary relatedness or if they have evolved independently.

The work presented here does, however, only achieves this goal partly. The proposed method remains vague as the description of it is very short. The intention of the method is applaudable, to aim from the beginning of a study to find the most suitable study organisms. However, as a user I would hope for some objective rules, so that I would feel confident that I have conducted a sufficiently robust search for possible study species. Instead, the author of the manuscript simply suggests to conduct an exhaustivity study, without defining exhaustivity. From my understanding, what is suggested is a thorough study, but the decision if it was exhaustive will be by made by the researcher themselves without specific criteria, and thus it would be a subjective decision.

Another problem with the manuscript is, that it is not written in a very scientific language. It is more an account of observations of the author, which would possibly fit a popular science magazine, but not a scientific journal that promotes quantitative plant sciences.

Additionally, the use of references is quite sparse and important statements are not supported. A few examples:

line 19: ... aesthetic biases are stronger than in the majority of other domains of research … (no reference)

line 49: theorized by Alan Turing in 1952 (no correct reference)

line 52: ... Shigeru Kondo was about to start a career in immunology ... (no reference)

line 55: Both Pomacanthus and Danio rerio were appreciated by the aquarists for their beauty ... (no reference)

Furthermore, it is common practice to use the last name of the first author (or both authors if only two contributed to the work) and the date to reference published work of other researchers in a manuscript text. Here, titles are used (Dr. and Pr., e. g. ll 48-52) and it seems that the most senior member of the lab is named, e.g. when an article from Nüsslein-Vollhard is mentioned it should instead have been Haffner (1996; https://doi.org/10.1007/s004270050051).

Where the author remarks on human perception of beauty (ll 82-86), this seems to belong into the field of Psychology and I feel that one should be reluctant to make those deductions without involving experts from that field.

I would have hoped that the introduction would present some overview on the topic of bias and possibly sampling strategies in (plant) sciences and to present the introduced method in the context of the common practices in the field. However, this is lacking.

The discussion is missing the critical consideration if the data base of the search, the data available online, is already biased towards aesthetically pleasing species. I assume that there is more information available for “beautiful” species than for “ugly” species. What impact such a bias would have on the suggested method is not sufficiently considered.

The manuscript needs a major rewrite. Clearer and more objective instructions are needed on how to conduct the SE method, so that it would be a tool for researchers to decide if they conducted a sufficiently rigorous search for study species. The logic on which the instructions are based should be explained. I also suggest to improve the language to make it more scientific and to follow the conventions of QPB for references.

---

## [Editor Report]

Dear Dr Galipot,

We have now received the comments from three reviewers for your manuscript. Again, I apologize for the very long delay in the process due the difficulties in finding reviewers as I wrote you before.

As you will see, the interest and originality of your point of view and proposed method are recognized, yet some clarifications and improvements are needed. Based on the reviewers comments I would suggest to resubmit the manuscript after rewriting and addressing the corrections asked in the comments. Notably it would be important to add an introduction on the topic of bias, clear examples/data and objectives rules, to make the method more convincing to potential users.

I thank you very much for your contribution to Quantitative Plant Biology.

Best regards

---

## [Reviewer Report]

This is my second review of this article.

I have seen with delight that the author has meticulously addressed all comments from all three reviewers in a very positive way.

All of my main issues with the article have been resolved. By choosing a better title, the focus of the article is now clearer, concentrating more on the exploration of biodiversity in colour pattern studies than in a quantifyable method to avoid bias.

The style of writing has also improved, being sufficiently scientific. However, I still feel that the English could be improved (to be more beautiful).

Some of the English style suggestions are:

l.8: remove the first ‘aesthetical’

l12: that (instead of which)

l29 replace ‘are needed’ by ‘have not or only rarely been performed’

l31 replace ‘to avoid’ by ‘reduce’

l62 what is a ‘fish pet’ ? replace by ‘a common species kept by aquarians’ or similar ?

l62: Kondo did not bifurcate, but his research did. Improve to ‘... led him to bifurcate his research interests ..’

l68: special issue (instead of exceptional issue)

l75 ‘research’ is missing after color pattern(s)

l79: remove ‘Pr.’

l82: The center of diversification is found in North America (remove ‘possess’)

l82: and Mimulus has genetically (instead of just ‘have’)

l86 and l87 remove ‘Dr.’

l88 by studying the role of ... (insert ‘studying’)

l90 rephrase: ... patterns, the lab investigated cats (and dogs) as candidates.

l91 insert before ‘companion’: ‘domesticated human companion species’

l91 remove ‘appreciated and’ as just ‘selected’ if fully sufficient here.

l97: would ‘we may change our taste preferences’ be better here than ‘reconsider our taste’ ?

l99/100: I suggest: To illustrate this with a personal example, initilly I have not been able to recognize the beauties of the common poplar tree. However, i have recently chanbed my perception after ...

l102/103: this sentence can go, as there is no further reference what thappened with the taste perception of the students.

l108: replace ‘last’ with ‘latter’

l110: replace ‘Nevertheless’ with something like ‘Surprisingly’

l111: replace ‘reveal’ with ‘trigger’

l112: replace ‘tendency’ with ‘preference’.

l126: replace ‘opinion on’ with ‘view fo’

l128 replace ‘colorless body’ (which would mean no colour) with ‘bland body color’ or similar.

l139, replace: ‘has inevitably changed for the better’ to: ‘has changed to a positive pereption’

l143: replace: ‘set my sights on’ by ‘considered’

l171: replace ‘describing’ by either ‘identifying’ or ‘finding’ or similar.

l172: replace ‘even’ by ‘including’

l175: Not sure what is meant with 'the impossibility to browse efficiently information."

l177: remove ‘must’ from ‘must require’

l180: rationalize (no end d)

l181: replace ‘personalized’ by ‘taylored’

l183: insert ‘extend of the’ natural biodiversity

184: replace ‘and because a look at’ by ‘as the consideration of ’

l185: replace ‘humans’ by ‘the researcher’

l185 strategic choices are required. (remove ‘needs to be done’).

l193: replace ‘as external phenotype’ by ‘an account of the optical appearance’.

l196: replace‘ Thanks to’ by ‘Following’

l198/199: ..., here are my personal criteria, additional to the classical criteria ...

l202: replace ‘a couple’ by ‘some’

l203: remove ‘animals’ (?)

l214/215: The timing of pattern formation could be crucial, if patterns only appear very early in development or in hardly accessible tissue.

l220: I don’t understand the worde ‘eventually’ here.

l222: it would be good to mention the refered too founding principle of evo-devo here.

l231 to 234 should be rephrased to be clearer and less extreme (e.g. undeniable).

l239 to 241 should be rephrased.

l244: remove ‘automatically’ and replace ‘precious’ by ‘useful’ or similar.

l246: replace ‘generate’ by ‘may generate’

l267: replace ‘biological’ by ‘biodiversity’

l308 to 310: Rephrase. What does ‘pinpoint’ mean. And what is meant by ‘classical exploration’ ?

l312: is ‘evident’ correct here? Rather ‘promising’?

l317: repace ‘IA might’ by ‘AI tools would’

---

## [Reviewer Report]

Many thanks to the author for all the improvements on the manuscript, which has been significantly modified and is clearly improved. The author’s choices are explained in his response to the reviewers.

However, I think that a few additional modifications are required to have the manuscript ready for publication, including some major and minor points.

Major points:

-Line 35: “These terms need a precise definition in the morphogenesis domain of research, which is one of the current works of a collaborative group of morphogeneticians”: Please provide a reference for this work (conference?), or, if a reference doesn’t exist, simply start more generally the next sentence, for instance: “Although a consensus on precise common definitions is probably still pending in the morphogenetic field, I define here a morphogenetic mechanism as a set….”

-Lines 43 to 52: this summary paragraph should start by “Here, …” (to clarify that it is a summary), and should be remodeled: in the current version, it gives more importance on the first part of the article which is actually short, in comparison with the rest of the article explaining the SE method. It would be better to balance the two parts in this summary, by shortening the first part, and elaborate a bit more on the SE method (its aims and virtues etc..).

-Line 54: I would remove “Discussion” section title, which seems awkward without a “result” section as in classical articles. I think this manuscript format would have sufficient structure based only on “introduction”, and then the sub-sections titles.

-Lines 102-158: the whole section (” The very changing nature of aesthetic judgments of species of interest”) is not easy to follow. To structure it more clearly, I would propose the following modifications:

Line 105: After the first sentence, add an introductory sentence, such as: “To illustrate the changing nature of our aesthetic judgments, I will choose several personal examples.”. Then the 3 examples can be described, starting, for each example, by explaining the main point, and then the illustrative situation: First (Line 106): the point “spending a long time in contact…” and the illustrative situation of poplar leaves. Second (Line 112/113): the point “changing scale” and the illustrative situation of discovering with students the Viburnum and Callicarpa colored fruits. Third (Line 137): the example of Sarcophaga.

-Line 150: “Galipot and Zalko, 2024, in prep”: this reference is not listed in the reference list, while I could see that a preprint was published.

In addition, the usage for preprints citation in the text is to mention “preprint” after the year. Please correct accordingly all along the text.

-Line 194 : “Some studies try to rationalized risk assessment (Forstmeier et al., 2017), but still are often too generalist to be able to provide personalized advice for your particular database.” : please reformulate this sentence as the meaning is unclear.

-Lines 196-204: this important paragraph regarding representativity/exhaustivity could be improved by giving an example, such as the process used in Galipot et al. 2023, which is nicely described with graphical representation in supplemental figure 3B of the preprint.

-Line 204: please explain briefly why beautiful species would be favored.

-Line 223: “, to use variation to better understand the processes (see non-mandatory step 3)”: this point is unclear, since it is not explained in step 3 how variation is used to better understand the process ?

-Line 264: Please provide an introduction sentence for the section.

-Line 269: please cite “Galipot, 2023” after “(Putative Growth Turing Color Patterns)”

-Line 302: Step 3: to balance the text, it would be nice to explain also briefly how Step 3 was used in the sudy Galipot et al 2023. (or alternatively, why it was not appropriated in that case?).

-Figure 4: Unfortunately, it seems that the iconography available does not convey clearly the ideas developed in the last section. From the current figure, I could not understand the “petits traits” technique. To simplify, I would suggest to drop the figure, and simply provide the curious readers with websites in the text. The website of Boulet is already cited in the text, and websites for Moebius/Jean Giraud can be added: for instance, Wikipedia (exists in english) and the official Jean Giraud website – where actually a nice example of “petit traits” drawing is visible: https://www.moebius.fr/page-Biographie.html ).

Minor points:

-Line 25: replace “like” by “for instance”.

-Line 26: add “, etc.” at the end of the list of patterns into parenthesis.

-Line 28: “with a preference of plant species when functional experiments are needed”: do you mean: ““with a preference for plant species where functional experiments are possible”?

-Line 34: add “for instance”: “(for instance, color patterns)”.

-Line 35: replace “synonym:” by “i.e.,”

-Line 82: replace “in the ENS de Paris” by « at the Ecole Normale Supérieure of Paris, »

-Line 82: “snobbism”: do you mean “reluctance”?

-Line 84: to clarify the sentence, replace “, are focused on them” by “, focus on them”.

-Line 89: to clarify the sentence, add “they are” before “colorful…”

-Line 157: to clarify the following sentence : “…, and led me to build and apply a method that I am now going to detail a little more.”, add “this” before “led”.

-Line192: “the possibility for the database to be open and collaborative (non-mandatory) would improve your database and the conclusions you draw from it”: it would be interesting to mention examples of citizen science-based databases that can be useful or that you have used?

-Line 245: add “e.g.,” before “color pattern”

-Line 246: “...if it is well done,”: could you please explain further “well done”? what are the important criteria?

-Line 271: “Before the study”: do you mean “Before our study” or “Before this study”?

-Line 282: To better introduce the paragraph, please cite Figure 3 and the preprint of the study, and precise that it refers to Step1 of the method:

For instance: “In the case of my study on checkerboard pattern (Figure 3, Galipot & Zalko, 2024 preprint), even if the biological survey was not as exhaustive as the precedent example, the step1 of the SE method permits to find very diverse examples of this quite rare pattern: …”.

-Line 291-293: “Finally, we selected a Lamiaceae plant, Scutellaria rubropunctata, which combined almost all of our criteria (Galipot and Tsukaya, in prep).”: Since the reference “Galipot & Tsukaya, in prep” is not a reference in the reference list, could you explain the criteria and provide a reference describing Scutellaria rubropunctata?

-Line 330: please correct the following sentence: “If typically there is a need of 10000 images to be trained (Lhermitte et al., 2022) and could be incompatible with the amount of species available before the study,..”. Do you mean: “If typically there is a need of 10000 images to trained a model (Lhermitte et al., 2022) and this could be incompatible with the amount of species available before the study,...” ?

---

## [Editor Report]

Dear Dr Galipot,

We have now received the reviewers comments on the new version of your manuscript. All the improvements were greatly appreciated by the reviewers and myself. Both reviewers ask however minor revisions. I recommend to follow their suggestions to finalize the manuscript.

Many thanks again for your contribution to QPB,

Looking forward to reading you.

Best regards

---

## [Reviewer Report]

Many thanks to the author for the revised version in which most of my comments were addressed, or a justification is provided in the author’s response for those comments which were not followed. Overall, I think the manuscript reads very well, expect a minor point of which must be corrected:

The complete reference for the preprint “Galipot and Zalko, 2024” should be added to the reference list, and the way it is cited in the text should be homogenized:

-Line 293: the reference is here correctly cited: e.i., “(Galipot and Zalko, 2024; preprint).”

-Line 297: the biorxiv link should be replaced by the reference’s citation as above.

-Lines 137-138: “ , see https://www.biorxiv.org/content/10.1101/2024.02.07.579346v2” should be removed.

-Line 276: “For the checkerboard pattern study…”: the reference should be cited here.

---

## [Editor Report]

Dear Dr Galipot,

Many thanks for your manuscript which appears almost ready for publication. All the reviewers comments were correctly addressed in this new version, except very minor points regarding a reference, which needs to be corrected before acceptance (see reviewer comment). Please provide a revised version including these minor corrections.

Thank you again for your contribution to Quantitative Plant Biology.

Best regards